# Three-Dimensional Stable Alginate-Nanocellulose Gels for Biomedical Applications: Towards Tunable Mechanical Properties and Cell Growing

**DOI:** 10.3390/nano9010078

**Published:** 2019-01-08

**Authors:** Priscila Siqueira, Éder Siqueira, Ana Elza de Lima, Gilberto Siqueira, Ana Delia Pinzón-Garcia, Ana Paula Lopes, Maria Esperanza Cortés Segura, Augusta Isaac, Fabiano Vargas Pereira, Vagner Roberto Botaro

**Affiliations:** 1REDEMAT, Federal University of Ouro Preto, Ouro Preto, Minas Gerais 35400-000, Brazil; prisiqueira2000@yahoo.com.br; 2Department of Chemistry, Federal University of Minas Gerais, Belo Horizonte, Minas Gerais 31270-901, Brazil; eder_siqueira@ymail.com (É.S.); anaelza015@gmail.com (A.E.d.L.); apfarmaceutica@gmail.com (A.D.P.-G.); anapaula.lopes@ctnano.com.br (A.P.L.); fabianovargas@yahoo.com (F.V.P.); 3Empa, Swiss Federal Laboratories for Materials Science and Technology, Applied Wood Materials Laboratory, 8600 Dübendorf, Switzerland; gilberto.siqueira@empa.ch; 4Faculty of Dentistry, Federal University of Minas Gerais, Belo Horizonte, Minas Gerais 31270-901, Brazil; mecortes@yahoo.com; 5Department of Metallurgical and Materials Engineering Federal, University of Minas Gerais, Belo Horizonte, Minas Gerais 31270-901, Brazil; augusta.cercearu@demet.ufmg.br; 6DFQM, Federal University of São Carlos, Sorocaba, São Paulo1 8052-780, Brazil

**Keywords:** alginate, cellulose nanocrystals, cellulose nanofibers, crosslinking, hydrogels, cell viability

## Abstract

Hydrogels have been studied as promising materials in different biomedical applications such as cell culture in tissue engineering or in wound healing. In this work, we synthesized different nanocellulose-alginate hydrogels containing cellulose nanocrystals, TEMPO-oxidized cellulose nanocrystals (CNCTs), cellulose nanofibers or TEMPO-oxidized cellulose nanofibers (CNFTs). The hydrogels were freeze-dried and named as gels. The nanocelluloses and the gels were characterized by different techniques such as Fourier-transform infrared spectroscopy (FTIR), scanning electron microscopy (SEM), transmission electron microscopy (TEM), thermogravimetric analysis (TGA), and dynamic mechanical thermal analysis (DMTA), while the biological features were characterized by cytotoxicity and cell growth assays. The addition of CNCTs or CNFTs in alginate gels contributed to the formation of porous structure (diameter of pores in the range between 40 and 150 μm). TEMPO-oxidized cellulose nanofibers have proven to play a crucial role in improving the dimensional stability of the samples when compared to the pure alginate gels, mainly after a thermal post-treatment of these gels containing 50 wt % of CNFT, which significantly increased the Ca^2+^ crosslinking density in the gel structure. The morphological characteristics, the mechanical properties, and the non-cytotoxic behavior of the CNFT-alginate gels improved bioadhesion, growth, and proliferation of the cells onto the gels. Thus, the alginate-nanocellulose gels might find applications in tissue engineering field, as for instance, in tissue repair or wound healing applications.

## 1. Introduction

Hydrogels, also commonly known as gels, are materials formed by a tridimensional polymer network able to absorb great amounts of water without dissolution [1]. They can be synthesized from synthetic polymer as polyethylene glycol (PEG), polyhydroxyethyl methacrylate (PHEMA) or from natural polymers as collagen, hyaluronic acid, and alginate [2]. In general, these gels present a low level of cytotoxicity and their high-water content gives them a structural similarity to an extracellular matrix (ECM) [3]. Therefore, such materials are of great interest to biomedical fields. Hydrogels, when compared to conventional cell culture substrates, present higher applicability and fundamental phenomena that regulate the cellular behaviors, such as spreading and directed differentiation of many kinds of cells [4]. They are also often used in the system of pharmaceutical drug release, present in various domains of medicine, including cardiology, oncology and, wound healing [5]. Nevertheless, physical characteristics of hydrogels such as dimensional stability, mechanical strength, degree of swelling, among others, are of high importance to their applicability. Their mechanical properties ensure physical stability [6,7] and they can also influence the cellular mechanotransduction, which have impacts in the proliferation, migration, and differentiation of cells, as stem cells [4,8].

Alginate, a polysaccharide synthesized from brown algae metabolism is often used in the formulation of gels. The alginate-based gels are of great relevance in the biomedical field due to their interesting capability to form hydrogels through ionic crosslinking mechanisms, where divalent cations, such as calcium ions (Ca^2+^), are often used. Alginate-gels present favorable characteristics to wrap cells, to regenerate tissues [4] and can also have self-healing properties [9]. However, generally they present low mechanical properties and dimensional stability [2,10], besides their limited cell attachment ability [4], which limits their use in tissue engineering applications including, for instance, bone, breast, and cardiac tissue repair or tumor fillings.

Alginate gels covalently crosslinked (chemical gels) present structures that are more mechanically stable [11] than the ones ionically crosslinked. However, the covalently crosslinked alginate gels can have some of their properties, such as biocompatibility, considerably compromised [12]. Some nanomaterials such as nanoclays, zeolites [13], and carbon nanotubes [2] have been studied as reinforcing agents in alginate-hydrogels, but they present some limitations for their applicability in the biomedical field due to their cytotoxicity.

Nanostructured hydrogels can simulate the complex interaction between cells and their environment to enable tissue growth and biological functions [14]. The combination of different types of nanocelluloses with various polymers allows potential applications of these nanostructured hydrogels in the biomedicine field, such as in drug release or as scaffolds for regenerative medicine [15]. It is important to emphasize the need of reinforcing hydrogels with nanomaterials, such as cellulose nanocrystals (CNCs) or cellulose nanofibers (CNFs), to improve their mechanical and biological properties for biomedical applications, mainly for bone tissue engineering [16]. For example, Park et al. [17] have developed a TEMPO-oxidized bacterial cellulose and alginate hydrogel (TOB) to encapsulate cells. They demonstrated that the addition of TOB was helpful to improve the mechanical properties and the chemical stability of the alginate hydrogels. In addition, increased cell proliferation was observed in the hydrogels containing nanocellulose when compared to pure alginate hydrogels. However, the use of cellulose produced by some types of bacteria, such as *Gluconacetobacter* is currently not economically competitive [18]. Thus, nanocelluloses obtained from wood have been reported by several authors as an interesting alternative in the preparation of alginate-based hydrogels for biomedical applications [7,16,19,20,21]. Using nanocellulose as a cytocompatible material, Lin et al. [22] developed alginate gels containing TEMPO-oxidized CNCs that presented higher mechanical strength than pure alginate gels. Aarstad et al. [23] studied the mechanical properties of the alginate and cellulose nanofiber (CNF/CNFT) gels and postulated the importance of the nanocelluloses’ surface charges on the mechanical properties of these materials. De France et al. [24] reported that the physical incorporation of CNCs increases the compressive modulus and the stability to degradation of alginate gels. Only little effect on the cell viability was observed after the addition of CNCs in the formulation of alginate-based gels, as described by Wang et al. [25]. In addition, several studies aiming to evaluate the cytotoxicity or genotoxicity of nanocelluloses obtained from wood have been developed [15,18,26,27,28,29]. However, it is important to postulate that the cellulose source, preparation procedures, and chemical modifications of nanocelluloses have direct influence on the physicochemical properties of the final products, thus affecting their cytotoxicity [30].

The presence of endotoxins after CNF preparation can cause inflammatory immune responses, mainly through susceptible immune cells. For further application of such materials in tissue regeneration or wound healing, nanocellulose endotoxin removal protocols should be considered. Nordli et al. [18] described the treatment of TEMPO-CNF with NaOH to eliminate endotoxins. These authors carried out extensive studies which indicated that TEMPO-CNF produced from bleached softwood pulp is not cytotoxic and that the endotoxin content was quite low (<50 endotoxin units/g). Similar conclusions were also recently reported by Chinga-Carrasco [15].

In this work, we evaluated the effect of thermally-induced crosslinking on the mechanical properties of alginate gels prepared by addition of different nanocelluloses and the correlation between their chemical composition and the morphology of the nanocelluloses (CNC or CNF) on their final properties. Since different types of nanocelluloses can be considered for the reinforcement of alginate-based gels for biomedical applications, in this work we hypothesized that a thermal post-treatment can increase the degree of crosslinking of the gels and consequently increasing mechanical strength, dimensional stability, cell growing, and bioadhesion. Thus, we evaluated the role of the additional crosslinking and the resultant mechanical properties of cellulose-based alginate gels on the cytocompatibility and cell growth. We finally demonstrate that these biomaterials present some physicochemical and mechanical properties that mimic the ones of living tissues. Moreover, to the best of our knowledge, this is the first time that a clear cell growth and bioadhesion is proved in nanocelluloses-alginate gels and combining mechanical properties and dimensional stability without compromising their porosity and capacity to absorb water and other biological fluids.

## 2. Experimental Section

### 2.1. Materials

All the cellulose fibers used in this work were obtained from bleached hardwood Kraft pulp from *Eucalyptus* (purchased from Bahia Pulp, Camaçari, Bahia, Brazil). For the oxidation treatment, 2,2,6,6-Tetramethyl-1-piperidinyloxyl (TEMPO), sodium bromide (NaBr ≥ 99%), sodium hypochlorite (NaClO) solution (10–15% chlorine), hydrochloric acid (HCl, 37%), and sulfuric acid (H_2_SO_4_, 98%) were purchased from Sigma-Aldrich (Belo Horizonte, Minas Gerais, Brazil). Sodium hydroxide (NaOH) was purchased from FMAIA S.A. (Belo Horizonte, Minas Gerais, Brazil) and used without any further purification. Sodium alginate (M_w_ = 1 × 10^6^ g·mol^−1^), a water-soluble copolymer of Manuronic (M) and Guluronic (G) acid units at M/G ratio = 1.56 and 250 cP of viscosity (2 wt % in H_2_O at 25 °C) was obtained from Sigma-Aldrich (data were provided by the manufacturer). Calcium chloride (CaCl_2_·2H_2_O) was purchased from Vetec Química Fina Ltda. (Duque de Caxias, Rio de Janeiro, Brazil). The phosphate saline solution was prepared with sodium chloride (NaCl), potassium chlorite (KCl), phosphate of anhydrous potassium (K_3_PO_4_), and sodium dodecyl sulfate (SDS) solution from SYNTH (Diadema, São Paulo, Brazil), monohydrated sodium phosphate (Na_3_PO_3_·1H_2_O) from Vetec Química Fina Ltda, and orthophosphoric acid (H_3_PO_4_, 85%) from NEON S.A. (Suzano, São Paulo, Brazil). All the reagents were used as received without any purification.

For the biological assays, mouse fibroblasts (L929) were purchased from American Type Culture Collection (ATCC: Manassas, VA, USA). Dulbecco’s Modified Eagle’s Medium High Glucose (DMEM), penicillin-streptomycin solution, and 3-(4,5-dimethylthiazolyl-2)-2,5-diphenyltetrazolium bromide (MTT) were purchased from Gibco-Invitrogen (São Paulo, São Paulo, Brazil).

### 2.2. Methods

#### 2.2.1. Extraction and Functionalization of the Nanocelluloses

##### Preparation of Cellulose Nanofibers

The cellulose nanofibers (CNF or CNFT) were produced by mechanical treatment using the Masuko grinder. First, the bleached cellulose fibers were dispersed in water up to a concentration of 2 wt % and let to swell at room temperature for 24 h. The suspension was further mixed using Ultra-Turrax (T50 basic, IKA^®^-WERKE, Staufen im Breisgau, Germany) for 5 min at 10,000 rpm. After homogeneous dispersion, the cellulose suspension was ground using a Supermass Colloider (MKZA10-20J CE Masuko Sangyo, Kawaguchi, Japan) equipped with non-porous grinding stones made out of SiC/Al_2_O_3_ particles in resin. The rotating grinding stone was driven at the nominal velocity (ca. 1500 rpm) to obtain the CNF suspension. The applied energy for the fibrillation was recorded and fixed at 2 kWh/kg of dry cellulose for the TEMPO-oxidized fibers and 10 kWh/kg of dry cellulose for the untreated fibers. The energy consumption of the process was recorded with a power meter coupled to the grinder machine.

##### Preparation of Cellulose Nanocrystals (CNC)

The cellulose nanocrystals (CNC) were prepared from bleached *Eucalyptus* pulp according to the method described elsewhere with minor modifications [31]. Acid hydrolysis with sulfuric acid (H_2_SO_4_ at 65% *v*/*v*) was performed at 50 °C under constant mechanical stirring. The amorphous regions of the cellulose fibers were preferably hydrolyzed, while the crystalline regions which have more resistance to the acid attack were kept intact. Successive washing steps with distilled water were carried out by centrifugation to remove the excess of acid (supernatant). The suspension was transferred to dialysis membranes. Successive water changes were made until pH ~6. The dialyzed materials were centrifuged obtaining suspensions of CNC at approximately 1.5 wt %.

Aiming to evaluate the influence of the morphologies of CNF and CNC as reinforcing agents in the alginate gels, the C6 hydroxyl groups of nanocelluloses were oxidized via TEMPO-catalyzed reaction. It was expected that the functionalized nanocelluloses could influence the final properties of the gels since the nanocelluloses are able to change their surface charges. The conversion of nanocelluloses’ OH groups to COOH groups was performed according to the method described by Saito et al. [32] using TEMPO as the catalyst and sodium hypochlorite as oxidant agent.

##### Preparation of TEMPO-Oxidized Cellulose Nanofibers (CNFTs)

The cellulose fibers were suspended in water to form a suspension at 2 wt % concentration. A stoichiometric relationship of 1.00 g of cellulose pulp, 0.016 g of TEMPO, 0.100 g of NaBr, and 5.35 mL of NaClO at 14 wt % was used. The reaction was carried out at pH 10 and at room temperature. The pH was adjusted before oxidation reaction with NaOH solution at 1.00 mol·L^−1^. After the oxidation, the fibers were washed with distilled water until the pH of the fibers was similar to the pH of the deionized water.

##### Preparation of TEMPO-Oxidized Cellulose Nanocrystals (CNCTs)

The TEMPO catalyst and the NaBr were dissolved in water until the concentrations of 0.1 mmol and 1.0 mmol per gram of CNC, respectively. Afterwards, they were mixed with the suspension of CNC at 2 wt %. The pH of the suspension was adjusted to 10 with the NaOH solution (0.25 mol·L^−1^). The reaction was started by the drop-wise addition of sodium hypochlorite (NaClO) solution. The volume of NaClO solution was chosen in a way to obtain a degree of oxidation of approximately 1.00 mmol per gram of the cellulose pulp. The CNCT suspension was purified using dialysis membrane of 6–8 kDa until its conductivity was similar to the distilled water of dialysis bath.

To determine the degree of oxidation (DO), it was performed a conductometric titration of the oxidized nanocellulose suspensions (CNCT and CNFT) as described by Saito & Isogai [33], (see Appendix A). The surface charge of CNC, CNCT, CNF, and CNFT suspensions at approximately 0.100 wt % concentration and pH = 7.00 were determined by Laser Doppler electrophoretic mobility measurements. The results represent a mean value of three samples at automatic mode (12 measurements for each sample). The Zeta potential (ζ) values were determined by the Nanosizer software (Malvern Instruments).

#### 2.2.2. Preparation of the Alginate and Nanocellulose Gels

In order to better evaluate the impact of the addition of nanocelluloses on the final properties of the alginate gels named as AlgCa, we prepared the pure alginate gel crosslinked with calcium ions. Based on an adapted method described by Lin et al. [22], we prepared a solution of the sodium alginate at 2 wt % which were transferred to Petri dishes and frozen with liquid nitrogen (−196 °C) for 5 min, followed by a freeze-drying step at −50 °C for 48 h. The freeze-dried materials were then added to a bath of CaCl_2_ at 2 wt % for 24 h to enable the crosslinking and the formation of the gels. Finally, the gels were washed with distilled water to remove calcium ions in excess, frozen with liquid nitrogen and freeze-dried again. To prepare the alginate nanocellulose-based gels, the nanocellulose suspensions (CNC or CNF) were dispersed in a solution of sodium alginate (2 wt %) by sonication for one minute. The concentrations of nanocellulose, described by the numbers in each sample name, are summarized in Table 1. The alginate-nanocellulose gels were prepared following the same procedure used to prepare the pure alginate gels.

#### 2.2.3. Determination of the Equilibrium Time and Moisture Uptake of the Gels

The equilibrium time (or stabilization time) of the studied gels was determined in milli-Q water (pH 6.2) and in phosphate buffered saline (PBS) solution (pH 7.4) at 25 and 37 °C. Dried gel samples were weighted and added to the PBS solution or milli-Q water. In regular time intervals, the samples were weighted. The removal of the excess of water from the samples’ surfaces was performed with filter paper. The maximum mass gain, corresponding to the moisture uptake, was calculated according to the Equation (1):(1)$$\mathrm{Moisture}\;\mathrm{uptake}=\frac{w_{t}{-w}_{0}}{w_{0}}$$
where: *w_t_* is the weight of the swollen sample at time *t*, and *w*_0_ is the initial weight of the sample before the swelling.

To determine the maximum swelling capability of alginate-nanocellulose gels, a similar procedure was used. In summary, the dried nanocellulose gel samples were weighed and added to PBS (pH 7.4) or milli-Q water (pH 6.2) at 37 °C for 2 h. The samples were dried, weighed, and the maximum moisture uptake (*w*/*w*) was calculated according to Equation (1).

#### 2.2.4. Cell Growth and Cytotoxicity Tests of Alginate and Nanocelluloses Gels

##### Cell Culture

Mouse fibroblast L929 were cultivated in DMEM high-glucose medium supplemented with 10% FBS and antibiotic solution (0.1 mg·mL^−1^ streptomycin and 100 U·mL^−1^ penicillin) according to the supplier instructions. The cells were grown at 37 °C in a humidified atmosphere of 95% air and 5% CO_2_. The cells were sub-cultured reaching 80% of confluence and seeded into 96-well plates at a density of 6 × 10^3^ cells/well and cultured for 24 h.

##### In Vitro Cytotoxicity Test

The cells death was measured using a dimethylthiazol diphenyl tetrazolium bromide (MTT) assay after 24, 48, and 96 h of treatment and evaluated by both direct contact assay and indirect contact assay in compliance with ISO-10993-5 (Biological evaluation of medical devices, Part 5: Tests for in vitro cytotoxicity) as related elsewhere [34,35]. The cell viability was expressed as a percentage detected for the treated cells (with gel contact) versus the viability of the cells incubated with culture assay medium which were used as a control (untreated). The data are expressed as the mean ± standard deviation (*n* = 6).

The direct contact assay used cells placed into each well of a 96-well plate at a density of 6 × 10^3^ cells/well and incubated for 24 h. The gels were cut into a piece of 5 × 5 × 1 mm^3^ and sterilized by UV-irradiation for 30 min. The gels were placed over a cell’s monolayer of fibroblast near confluence and incubated for 96 h. Samples were evaluated at 24, 48, 72, and 96 h. Subsequently, at each time point, the gels and the cell culture medium were removed, and the wells were washed with sterile phosphate buffered saline (PBS) solution. Afterward 10% MTT solution was added to each well and after 4 h the formed salts were solubilized into formazan by adding 1% sodium dodecyl sulfate (SDS) solution, and the absorbance of the content of each well was measured at 570 nm using a Thermo Scientific Multiskan Spectrum MCC/340 spectrophotometer.

The indirect contact assay or elution test was performed by using the gels extractable from cell culture medium. The gel’s extract was prepared by soaking the gels in freshly prepared DMEM high glucose culture medium of composition as described above for 48 h and subsequently filtered through a 0.22-mm syringe filter. The extractable from the samples that migrated into the nutrient medium was placed into a layer of cells and incubated for 24, 48, 72, and 96 h. Subsequently, at each time point, the cell culture medium was removed, and the MTT test was performed in a similar way as related before.

##### Cell Proliferation and Bioadhesion

The proliferation and bioadhesion of cells in the gels was evaluated by SEM imaging. The cells were seeded on gel samples. Gels were cut into a piece of 5 × 5 × 1 mm^3^ and sterilized by UV-irradiation for 30 min. Prior to cell seeding, gels were immersed in FBS for 1 h and then seeded with fibroblast cell suspension at a density of 5 × 10^6^ cells/mL and kept for incubation at 37 °C in CO_2_ atmosphere for 7 days. At the end of the incubation period, the gels containing cells were washed carefully with PBS and fixed with 2.5 wt % glutaraldehyde at 37 °C. The dried samples were then immersed in graduate series of gradient ethanol (10, 20, 40, 60, 80, and 100%) and then vacuum-dried in a desiccator. The samples were turned conductive by sputtering with an ultra-thin gold coating at a very low deposition rate and analyzed using a scanning electron microscope (Quanta FIB EGF 3D FEI).

#### 2.2.5. Characterization Methods

Images of the oxidized and bare nanocelluloses were obtained by transmission electron microscopy (TEM) in an electronic microscope Tecnai G2-12-SpiritBiotwin FEI-120. The samples were prepared through the deposition of 0.01 wt % nanocellulose suspensions previously dispersed in grids of copper covered with Formwar film.

The infrared spectra were recorded at room temperature in a spectrometer Perkin-Elmer Spectrum Version 10.03.02, in wavelengths ranging from 4000 to 500 cm^−1^, 2 cm^−1^ resolution, and accumulation of 20 scans.

To determine the degree of crystallinity (C_IR_) of nanocelluloses a Shimadzu diffractometer X-ray diffraction (XRD)-model 6000, monochrome Cu K-α radiation (λ = 0.15428 nm), 30 kV voltage, and current of 30 mA was used. The measurements were carried out in continuous scan mode with speed of 2°·min^−1^ in the Bragg angle (2θ) between 5° and 50°. The C_IR_ values were calculated according to the classical equation proposed by Segal et al. [36]:(2)$$C_{\mathrm{IR}}\left(\%\right)=\frac{I_{200}{-I}_{am}}{I_{200}}\times 100$$
where *I*_200_ is the maximum intensity of the *I*_200_ plan diffraction (200) attributed to crystalline region of cellulose and *I_am_* is the diffraction intensity in 18° (Bragg angle) which is attributed to the amorphous region of cellulose.

The microstructures of the gels were analyzed by scanning electron microscopy (SEM) using a scanning electron microscope FEI Quanta Focused Ion Beam systems (FIB) EGF 3D. The samples were previously cryo-fractured in liquid nitrogen. Samples were fixed on the SEM supports and coated with a continuous film of carbon of approximately 15 nm thickness. Scanning electron micrographs were also used to estimate the pore sizes of the neat alginate and nanocellulose-alginate gels by analyses of the micrographs with the software Image J. A minimum of 15 measurements per sample was performed.

The thermogravimetric analyses were carried out in a TGA-DTG-60 Shimadzu in the range between 25 and 600 °C, heating ratio of 10 °C·min^−1^, and under N_2_ atmosphere.

Dynamic mechanical thermal analyses (DMTAs) of alginate and nanocellulose-alginate gels were carried out in a DMA analyzer (Netzsch-242 model). The measurements were performed at compression mode in the temperature range between 20 and 70 °C, 3 °C·min^−1^ heating rate, 1 Hz frequency, and 5 N oscillating load. To determine the influences of thermal post-treatment on mechanical properties of gels we carried out two kinds of trials. For the first one two consecutive heating/cooling cycles were performed on the same sample in the furnace of DMTA apparatus to determine storage modulus (E’_1_ and E’_2_: storage modulus determined during first and second heating/cooling cycles, respectively). For the second one the gels were thermally post-treated in an oven at 70 °C during 1 or 4 h. All measurements were performed at duplicate.

## 3. Results and Discussion

### 3.1. Morphological, Structural, and Chemical Properties of Nanocelluloses

The surface charge, the morphological characteristics and the crystallinity index of the nanocelluloses have a direct influence in the physicochemical and biological properties of the alginate gels. In the present study, we determine the surface charge and the structural features of the nanocelluloses (CNC or CNF) by ζ potential measurements and micrographs obtained by TEM, respectively.

The CNC and CNF presented a negative ζ potential value of −43.5 ± 5.91 mV and −28.8 ± 6.22 mV, respectively. These negative surface charges were due to presence of ester sulfate groups on CNCs imparted after acid hydrolysis and residual carboxylate groups of uronic acids (hemicelluloses) on CNFs (see Appendix A). After TEMPO-mediated oxidation reaction, these values increased towards −57.1 ± 6.41 mV and −69.6 ± 9.10 mV for CNCT and CNFT, respectively. As expected, this behavior was due to an increase of carboxylate groups on CNCT and CNFT after oxidation reaction. 

Morphological structures of nanocelluloses are shown on the micrographs presented in Figure 1a–d. The CNC exhibit a typical rod-like morphology with an average length of 170 ± 15 nm and average diameter of 4 ± 1 nm. After TEMPO-mediated oxidation (CNCT), significant changes were not observed in the morphology of the CNCs, as already reported in the literature [7,37]. Due to limitations of the grinder technology used to produce cellulose nanofibers, it was possible to have residual cellulose fibers in the nanofiber suspensions even applying 10 kW·h/kg of cellulose in the extraction process. However, the microscopy analyses revealed that the majority of the materials were fibrillated. The agglomerates observed in the CNC samples were due to lateral interactions between CNC and the presence of hydroxyl groups on the CNC surface, and also due to the high specific surface area of the CNC. Nevertheless, in Figure 1c,d, it is possible to observe that the CNF present a higher degree of entanglement and/or aggregation when compared to the CNC. This difference is ascribed to the higher aspect ratio (L/D) of the CNF which present an average diameter between 30 and 50 nm and 2–5 µm in length. This result is similar to that described by Dufresne [38]. After TEMPO-catalyzed oxidation, CNFT had their lateral dimensions decreased (average diameter between 5 and 10 nm), when compared to unmodified CNF, but keep their length in the micrometer scale. This result is in agreement with the ones reported by Sehaqui et al. [39].

To highlight the oxidation of the primary hydroxyl groups on the CNF and CNC (through the TEMPO-mediated oxidation reaction) in Figure 1e is presented in the FTIR spectra. The appearance of the band attributed to symmetric stretching vibration of carboxylate groups (COO^−^) at approximately 1604 cm^−1^ confirms the oxidation reaction. There is an asymmetric stretching at 1411 cm^−1^, which is also attributed to the COO^−^ groups. A decrease of the intensity of the band related to −OH groups stretching vibrations at around 3300 cm^−1^ was also observed. The spectra of the CNF are very similar to the spectra of CNC: they present a band at 3300 cm^−1^ attributed to the stretching vibrations of −OH groups; at 2900 cm^−1^ there is the vibration bands of C–H groups and the deformation of methyl groups (CH_2_) at around 1430 cm^−1^; the absorptions bands at 1200 and 920 cm^−1^ are attributed to the polysaccharide structure of cellulose [40]. The absorption bands of sulfate ester groups (imparted on CNC after the hydrolysis with sulfuric acid) are, in general, less intense and no characteristic bands are observed in the spectra [41].

The XRD analyses of the nanocellulose (Figure 1f) show that the chemical functionalization of the nanocellulose surfaces preserved their original morphology and kept the integrity of the nanocrystals. For all studied nanocelluloses, two main diffraction peaks characteristic of the cellulose type I were observed: peak of maximum intensity at 2θ = 22.5°, which corresponds to the crystallographic plan (200), and at 2θ = 15.3° that are attributed to the plan (110). One can observe a shoulder located at around 2θ = 21°, more intense for the CNCT samples, which represents a reflection of the plans (102/012). A well-defined peak is also observed at 2θ = 34.5°, attributed to the plan (040) [42]. This crystalline structure corresponds to two parallel glycosidic chains, which have been referred to as the cellulose polymorph of greater mechanical strength [43]. As expected, the crystallinity index (C_IR_) determined for CNC (90%) and CNCT (82%) were higher than the ones found for the bare CNF (62%) and CNFT (60%) due to the removal of the amorphous regions of the cellulose during acid hydrolysis [44]. Thus, it was demonstrated the preservation of the nanocellulose morphologies and their crystalline integrity after the TEMPO-mediated oxidation reactions. Finally, the degree of oxidation (DO) or total content of carboxylic groups present on the TEMPO-oxidized nanocelluloses was determined by conductometric titration based on the method proposed by Saito and Isogai [33]. The results have shown a DO of 1.210 mmol·g^−1^ cellulose for CNFT and 0.513 mmol·g^−1^ cellulose for CNCT (see curves at Appendix A). A lower degree of oxidation found for CNCT when compared to the CNFT is expected due to the lower reactivity of the crystalline regions of the CNC [33,45]. It is suggested for CNFT samples that a considerable percentage of primary hydroxyl groups on their surface are accessible during the TEMPO-mediated oxidation reaction [32].

### 3.2. Pure Alginate and Nanocellulose-Alginate Gels

Crosslinked pure alginate gels (AlgCa) with calcium ions present high porosity and are known for their healing and anti-tumor potential applications [46,47]. However, they are unstable in physiological pH or in environments with high concentration of phosphate or citrate ions because Ca^2+^ ions can be extracted from the hydrogel structure leading to a collapse [46]. This study was developed aiming to improve the mechanical properties and dimensional stability of alginate-based gels without compromising the biological features of the materials. Thus, TEMPO-oxidized and unmodified cellulose nanofibers and cellulose nanocrystals have been added to alginate solutions (2 wt %) to prepare potential materials for biomedical applications.

Figure 2a shows the ionic crosslinking mechanism of alginate gels with Ca^2+^ ions and the influence of concentration and different types of nanocelluloses on the dimensional stability of the gels. Unmodified and modified cellulose nanofibers (CNF and CNFT) which are longer (micrometer scale) and more flexible than the CNCs are able to form a more entangled network in the material influencing positively the structural properties of the gels [24]. Moreover, the CNFT, which has carboxylate groups on their surface can participate actively in the crosslinking structure of the gels and also in the structure of them, imparting higher dimensional stability to the materials. The CNC/CNCT stands out by favoring the gelation mechanism and increasing the mechanical properties of the final materials [48,49]. The intense network of hydrogen bonds established between the CNC/CNC in combination to their morphologies contributes to forms a more fragile and brittle material when compared to the gels prepared with CNF. Taking into account that the dimensional stability of the gels changes linearly with the concentration of the reinforcement agents (CNC/CNCT or CNF/CNFT), alginate gels with the lowest CNC or CNF contents (10 wt %) presented heterogeneous structure, and lower dimensional stability when compared to the gels containing greater amount of nanocellulose concentrations (36 or 50 wt %).

Crosslinking reactions are generally used to form polymeric networks with improved physicochemical and mechanical properties. To confirm physical interactions between the alginate chains with the calcium ions and the nanocelluloses in our systems we carried out analyses of infrared spectroscopy. The FTIR-ATR spectra of alginate (AlgCa) and alginate-nanocellulose gels (AlgCNC36; AlgCNCT36; AlgCNF36; AlgCNFT36) presented O–H stretching vibration band in the range from 3600 up to 3200 cm^−1^ and C–H stretching at around 2900 cm^−1^. Absorption bands at 1594 and 1410 cm^−1^ are characteristics of asymmetric and symmetrical elongations of carboxylate groups of the alginate polymer, respectively [50]. The band of symmetrical deformation of the COO^−^ group at 1407 cm^−1^ for the neat polymer (alginate) exhibited a displacement towards 1424 cm^−1^ when calcium ions were added into alginate solutions. This behavior is an evidence of the ionic crosslinking of the carboxylate groups present in the system. According to Sartori et al. [51] the substitution of sodium ions by calcium can modify the load density, atomic mass, and the radius around the carboxyl groups and a displacement of characteristic bands in FTIR spectra. In addition, the displacement towards a higher wavenumber can also be attributed to the intermolecular interactions between the COO^−^ of alginate and the nanocelluloses to form a crosslinked network. An increase of the transmittance intensity around 3600 up to 3200 cm^−1^ is ascribed as hydrogen bonds formed between alginate and nanocelluloses. The bands between 800 and 1200 cm^−1^ are attributed to the polysaccharide structure (C–H, O–H, C–O and C–O–C stretching vibrations in the glycosidic ring). Therefore, due to the displacement of characteristic bands, the electrostatic interactions between the COO^−^ groups present in the system and the Ca^2+^ ions can be confirmed. Thus, the FTIR spectra also suggest the intermolecular interactions between the polymer chains of alginate and nanocelluloses.

The pore size of the gels and their distribution are very important for their capability to absorb liquids and fluids [12]. These properties will impact directly the cell growth within such materials as they play a role on the transport of nutrients and other biological fluids. To demonstrate how the morphological properties of the alginate-based gels are influenced by the addition of nanocelluloses on their composition, we performed SEM of the samples. Figure 3a–c shows the SEM micrographs of the cross-section of the gels cryo-fracturated in liquid nitrogen (N_2_) and Appendix A presents the pore sizes and standard mean deviation values. The micrographs present a typical morphology of freeze-dried materials [52] with a high porosity. As observed in the images of Figure 3a–c, the addition of TEMPO-oxidized nanocelluloses (CNCTs and CNFTs) apparently contributed to the formation of pores with higher diameter (ranging from 40 to 150 μm) when compared to pure crosslinked alginate gels (pore diameters of 20 up to 40 μm). Additionally, there is not apparent aggregation between nanocelluloses and the gel structure is more able to form a web-like structure with larger pores when compared with alginate gels prepared without addition of nanocelluloses. These porous structure and pore diameters are described as good and favorable for cell growth [53,54]. Scanning electron microscopy images of the alginate-gels prepared with unmodified nanocelluloses are also presented in Appendix A.

To evaluate the influence of the morphology of the gels in their physicochemical and biological properties, the maximum swelling capability of the materials in different conditions (e.g., in milli-Q water at pH 6.2 and PBS at pH 7.4, and at 25 and 37 °C) was studied. First, the swelling kinetics trials of the calcium crosslinked alginate-gel (AlgCa) were carried out (Figure 3d) in milli-Q water. In such experiments, approximately 60 min were needed to obtain a maximum and constant weight increase of 7.0 g/g. In PBS medium, the swelling equilibrium was reached after 30 min in pH 7.4. These materials presented a maximum weight increase of approximately 10 g/g at 25 and 37 °C. The small variation in the degree of swelling of the calcium-alginate gels as a function of the temperature in PBS can be explained by the presence of hydrophilic groups that at higher temperatures have favored the polymer-water interactions. It is well known that the swelling depends on the chemical composition of the polymers used to form the gel structure, on the pH and on the ionic strength of the medium in which the gels are immersed. When the gel hydrates, the polymer chains interact with the solvent molecules and swell. The retraction and expansion of the gels occurs until the equilibrium is reached at a specific temperature [55,56]. In addition, alginate ionically crosslinked gels without nanocelluloses addition are not stable in physiological medium [57] and they collapse quickly in the presence of chelating agents for calcium ions (e.g., phosphates) or monovalent ions such as Na^+^ or K^+^ [12]. This explains the greater and faster swelling of the alginate gels in PBS at pH 7.4, which presented an equilibrium time of 30 min and maximum weight increase of 10 g/g compared to their swelling in milli-Q water where the equilibrium time was of 60 min and their maximum weight increase of 7 g/g. Therefore, to obtain the maximum degree of swelling for alginate-nanocellulose gels and to understand the behavior of these materials in physiological medium for future biomedical applications, we performed new swelling tests in milli-Q water (pH 6.2) and in PBS medium (pH 7.4) at the temperature of 37 °C for 2 h. One can clearly observe in Table 2 that the established time interval of 2 h for these tests is higher than the time needed by the gels to reach swelling equilibrium. It can be confirmed the influence of the nanocellulose morphologies (nanocrystals versus nanofibers), chemical functionalization (e.g., TEMPO-oxidation) and pore sizes of gels on the degree of swelling. The oxidation of cellulose fibers results in higher water content without significant changes in the morphology of the gels [32].

The data presented in Table 2 show that cellulose nanofibers are more able to influence the structural properties of the gels and allow higher moisture uptake when compared to the gels prepared with similar cellulose nanocrystals contents. This fact is attributed to the higher aspect ratio of the nanofibers (evidenced by TEM) and pore sizes (SEM images and Appendix A). The cellulose nanofibers form an entangled network that more effectively participates on the structural of the gels. In addition, the presence of carboxylate groups on the surface of the modified cellulose nanofibers (CNFTs) allows them to contribute to the crosslinking [58]. Thus, the CNFTs form a semi-interpenetrating polymer network (semi-IPN) with the alginate gels, as reported by Lin et al. [22]. As a result, one can observe a significant increase of moisture uptake up to 18 g/g for the gels containing CNFs or CNFTs. The addition of CNCs and CNCTs into the alginate gels did not influence their degree of swelling when compared to the materials prepared without nanocellulose (AlgCa).

In this case, cellulose nanofibers can be used for synthesis of alginate-based gels acting as “support” for the polymer network. They seem to help the opening of the gel structures due to their entanglement and enhance the swelling capability of the material. It is known that the swelling capability of gels is a result of the diffusion of solvent molecules into the gels and of the interactions between water molecules and polymer structure. These molecules’ retention process initially occurs in the empty spaces of the gels. However, swelling inevitably involves changes in the spatial arrangement of the polymeric chains and results an increase of the volume of the sample. After immersion the CNC and CNCT alginate-gels for 2 h in PBS (pH 7.4) at 37 °C, it was observed that the samples disintegrate. The samples containing CNFs or CNFTs showed higher resistance against collapse and solubilization meaning that they were more mechanically stable in PBS medium. For the last one system, only after 22 h in the PBS medium was it possible to observe a slight weight loss. The stability of the gel is important because it is directly related to its performance. Studies describing the crosslinking kinetics through calcium bridges were performed by Lee and Mooney [59]. Their results showed that slower kinetics allows to obtain gels with more homogeneous structure and increased mechanical properties. Sartori et al. [51] also studied the concentration of crosslinking solution and Zhang et al. [12] the IPN formation between alginate chains to decrease the degradation kinetics in physiological medium. In the present study, it was observed that the addition of the CNFs/CNFTs slightly increased the porous size of alginate-gels (as previously discussed based on SEM micrographs and Appendix A) and their stability in physiological conditions (PBS pH 7.4 at 37 °C).

We carried out thermogravimetric analyses of the alginate-gels to take some information about thermal stability of the materials based on their compositions (with and without nanocelluloses). Based on thermogravimetric curves presented in Figure 4 and in Appendix A, it is possible to compare different behaviors of the gels and to analyze how the effects of treatments as the crosslinking density or oxidation of nanocelluloses influences the final properties of the gels. The thermal degradation curves (TGA) of pure alginate-gels (AlgCa) and alginate-nanocellulose gels (AlgCNC36; AlgCNCT36; AlgCNF36; AlgCNFT36) present three major mass loss events. The first one (from 20 to 200 °C) corresponds to the loss of water and other volatile components present in the materials. At this initial stage the first decomposition of the oligosaccharides take place [60,61]. It can also be observed from the first event a lower thermal stability for alginate-gel without nanocelluloses addition (AlgCa). The second one (between 200 and 425 °C) is attributed to the complex degradation of glycoside chains present in the structure of alginate-nanocellulose gels: depolymerization, dehydration, and decomposition of the glycosidic units. At this event, there was a mass loss of approximately 55% up to 350 °C for alginate gel (AlgCa), 375 °C and 400 °C for alginate/CNFT gel and alginate/CNF, respectively (AlgCNFT36; AlgCNF36) and up to 400 °C and 425 °C for CNCT/alginate and CNC/alginate gels, respectively (AlgCNCT36; AlgCNC36). This phenomenon confirms that nanocelluloses participate actively in the thermal stability of alginate-gels and a displacement of the temperature of thermal degradation for gels prepared with oxidized nanocelluloses was observed. Besides, the hemicelluloses present in the CNF structure also promote changes in the initial degradation temperatures of cellulose nanofibers-based gels (AlgCNF36 and AlgCNFT36). The third one occurs due to the degradation of the carbon-based residues and to the formation of volatile products of low molar weight. According to Han et al. [62], the residue from the degradation process can be attributed to the electrostatic interaction between alginate and nanocelluloses in Ca^2+^ ions medium, resulting in a strong thermal stability mainly observed in the AlgCNCT36 and AlgCNFT36 samples, as indicated in Figure 4. When evaluating the influence of the nanocellulose contents in the thermal stability of alginate gels (Appendix A), it can be observed, once again that a lower concentration of nanocelluloses (10 wt %) in the alginate gels has little contribution on their final properties. This fact is less evident for gels containing CNFs depending on the lower dispersion of the unmodified nanocelluloses in the alginate matrix.

The mechanical properties of the gels are of great importance to their potential final applications. For their use as implants, E-moduli ranging 0.100 up to 1.00 MPa, tensile strength of 1.00 to 10.0 MPa, and compressive strength between 20 and 60 MPa are required because they are comparable to some human tissues such as tendons and dermis [6]. To correlate the macroscopic properties of the gels with thermally-induced molecular relaxation and physicochemical interactions in the crosslinked alginate gels and nanocellulose-alginate gels, we carried out dynamic mechanical thermal analysis (DMTA).

The changes of storage moduli (E’) and tan δ (ratio between loss and storage modulus) during the first (E’_1_, tan δ_1_) and second (E’_2_, tan δ_2_) heating cycles are presented in the Figure 5a,b for the neat alginate gel. An increase of approximately 94% of the E’ at 37 °C can be observed (Figure 5a), when the first and second heating cycles are compared. The E’ values increases from E’_1_ = 4.95 MPa to E’_2_ = 9.60 MPa. For higher temperatures (70 °C), this difference between E’_1_ and E’_2_ decreases to about 22%. We hypothesize that this behavior is due to an additional ionic crosslinking through Ca^2+^ bridges induced by a thermal post-treatment during heating/cooling cycles of the alginate gels without nanocelluloses addition.

Such observation is supported by the disappearance of tan δ peak (Figure 5b) in the second heating cycle (temperature range between 30 and 80 °C). The relaxation of the polymer chains occurs in the glass transition region and results from the molecular movements over long distances, which involve the segments of the main polymer chains [63]. Thus, the increase of ionic crosslinking density decreases polymer chains mobility [64].

To understand the influence of the nanocelluloses concentration on the thermomechanical properties of alginate gels, we prepared materials with two different concentrations (36 and 50 wt %) and carried out DMTA during two heating cycles. For this part of our study, the E’ was evaluated (after the first and second heating cycle, respectively, E’_1_ and E’_2_) in the temperature range (from 35 up to 42 °C) as indicated in Table 3.

Alginate and CNC/CNCT gels did not show significant changes of E’ in the second heating cycle. As shown in Table 3, an increase in the mass ratio of CNC or CNCT (from 36 wt % to 50 wt %) in the alginate gels decreased their storage moduli. This can be explained by the formation of phases and/or increase of gels pore sizes when increasing the nanocellulose contents in the gels. The lateral interactions between the cellulose nanocrystals (CNC–CNC or CNCT–CNCT) increase when their concentration also increases in the gel leading to a poor dispersion of the CNC and to the formation of phases. This phenomenon is observed by the decrease of the storage modulus values in the whole temperature range used. The lower values of E’ for AlgCNCT when compared to AlgCNC gels may be justified by the possible influence of the CNCT (with higher carboxylic group contents) and competition for the calcium ions with the chains of the alginate matrix. The addition of CNCT may broke the “egg box” structure of the material and impact their mechanical properties. It is therefore emphasized that the cellulose nanocrystal-alginate gels (AlgCNC and AlgCNCT) were more rigid and presented better mechanical properties (specifically in terms of E’) when compared to the corresponding cellulose nanofibers-alginate gels (AlgCNF and AlgCNFT). De France et al. [24] reported that the addition of CNC in alginate gels increases the compressive modulus and they have little effect on cell viability [25]. For CNF-alginate gels, a decrease in the storage moduli was observed when the nanocelluloses content increased. However, when TEMPO-oxidized cellulose nanofibers (CNFTs) were used for gel preparation, we observed an increase in the storage modulus when the nanocellulose content also increased and in the second heating cycle. This behavior was opposite to the one observed for the CNCT-alginate gel. It can be justified by the increase of carboxylic groups on nanofibers after TEMPO-catalyzed oxidation reaction that allows a better thermally-induced interaction between alginate and CNFT through ionic crosslinking with Ca^2+^ ions. As a consequence, we observed clear improvements in the mechanical properties of the CNFT-alginate gels similar to the results reported by Lin et al. [22].

Therefore, our findings indicate that the CNFTs participate more actively in the micro-structure of alginate gels through the formation of semi-IPN gels, which directly impact the values of E’. To obtain improvements in the mechanical properties of the CNFT-alginate gels and corroborating with the results obtained from the analyses of DMTA (Table 3), a thermal post-treatment in an oven at 60 °C for the AlgCNFT50 gels was performed (see Figure 5c). A considerable increase of E’ value (in the temperature range studied) can be observed when longer times of thermal post-treatment were carried out. This contributes to a possible additional thermally-induced ionic crosslinking between the Ca^2+^ ions and the carboxylic groups available in the structure of the nanofibers. The TEMPO-oxidized cellulose nanofibers-alginate gels (AlgCNFT), mainly due to the carboxylic groups find on the CNFT and their ability to form entangled and flexible networks, presented improved mechanical properties after relatively short periods (4 h) of thermally-induced post-treatment. Such gels are considered, therefore, as promising materials for biomedical applications.

### 3.3. Cytotoxicity and Cell Growth Tests

To demonstrate the great potential of the nanocellulose-alginate gels as biomaterials, we chose the samples at 50 wt % concentration of TEMPO-oxidized cellulose nanocrystals or nanofibers to carry out cytocompatibility tests. The same tests were carried out on the alginate-gels without nanocelluloses to compare the results. Cell viability at 24, 48, 72, and 96 h after the direct and indirect contact of AlgCa, AlgCNCT50 and, AlgCNFT50 gels on L929 fibroblast cells are shown in Figure 6a,b, respectively.

For analyses of these tests including cytotoxicity in vitro, we adopted the following criteria of measurements based on cell viability when compared to the control: non-cytotoxic, >90% cell viability, slightly cytotoxic = between 60% and 90% cell viability; moderately cytotoxic between 30% and 59% cell viability; and severely cytotoxic, ≤30% cell viability.

Our results demonstrate that the direct contact test of AlgCa, AlgCNCT50, and AlgCNFT50 gels on L929 fibroblast cells show moderate cytotoxicity effect (range 60–85% of viability) and the materials that showed better cytocompatibility profile were AlgCNCT50 and AlgCNFT50 (Figure 6a). At the same time, it can be seen that the cell growth remains stable within the time culture for AlgCNCT50 and AlgCNFT50 gels. The cell population was higher on the AlgCNCT50 and AlgCNFT50 than on the pure alginate-gel (AlgCa) after two and four days of culture, especially for the AlgCNCT50. In this way, direct contact exposure of the biomaterials to the cultured cells simulates in vivo conditions of direct interactions of transplanted or host cells with implanted biomaterials and allows in vitro evaluation of cell-biomaterial interactions [34]. Moreover, the addition of CNC and CNF can promote the cell attachment, spreading and growth on the alginate gels in accordance with the previously established by Naseri et al. [7] and Domingues et al. [35].

It is important to highlight that direct contact of AlgCa and AlgCNCT50 gels at 24 h decreased the cell viability. This behavior can be attributed to the high swelling of the gel with the culture medium and the increase of Ca^2+^ ions that reduce the water content available in culture medium and increase the toxicity [65]. However, after 48 h of contact with cells the pure alginate gel was completely disintegrated and the culture medium was fully available to the cells. The culture medium was favorable to cell proliferation up to 96 h of contact.

For the indirect assays (elution tests), the extractable materials were used to evaluate the effects of possible cytotoxic contaminants that may readily be extracted from the biomaterials [34]. It was observed that AlgCa, AlgCNCT50, and AlgCNFT50 gels eluates showed non-cytotoxic to moderate cytotoxic activity (75–98% of viability) and showed impressive cytocompatibility profile (Figure 6b).

Although all materials present good cytocompatibility with L929 fibroblasts cells, the AlgCa and AlgCNCT50 gels unfortunately lost their physical integrity and were disintegrated in the culture medium at the first contact time. For that reason, further investigations are needed to consider these materials as interesting substrates for tissue engineering or cell growth supports. On the other hand, the gels containing TEMPO-oxidized cellulose nanofibers (AlgCNFT50) nicely maintained their dimensional stability and presented the best compatibility conditions for adhesion and growth of fibroblasts on their surface due to the presence of entangled nanofiber network.

Therefore, we focused our investigations in the gels containing TEMPO-oxidized cellulose nanofibers (AlgCNFT50). Scanning electron micrographs of L929 fibroblast seeded on AlgCNFT50 gels after two (Figure 7a,b), four (Figure 7c,d), and seven days of incubation (Figure 7e,f) revealed that the fibroblasts were immobilized within the pores of the alginate gel, but they were also found on the nanofibers (Figure 7c,d) and on the surface of the gels (Figure 7a,b). Besides, the morphology of the fibroblasts was kept spherical and at extended shape usually found in monolayer culture on gel surface. The spherical-shaped cells found on these gels’ scaffolds after seven days of incubation (Figure 7e,f) can be considered as a strong indication that the cells were growing and had the capability of proliferation after these time intervals of incubation [7]. Rashad et al. [14] associated the spherical shape of the fibroblast cells in their hydrogels, with the physicochemical properties of the material such as the presence of carboxymethyl surface group and/or the hydrogel stiffness. They observed that the TEMPO-oxidized nanocellulose-based hydrogels kept the elongated spindle-shaped morphology characteristic of fibroblasts. They concluded that TEMPO-CNF hydrogels were more promising material for tissue engineering applications.

It is interesting to observe, from a closer view of the L929 fibroblast cells on the gels presenting rough surface textures (Figure 7b), that rope-like filaments were growing out of the cells. These filaments might help cells migrate through the porous surfaces and promote interaction between the cells [65]. It is well known that the pore size of scaffolds affects cell bioadhesion, proliferation, and differentiation, and gel pore size must be small enough to ensure their mechanical integrity; however, they must remain large enough to allow cell growth and nutrient diffusion within the tissue [66].

In Figure 7e,f, AlgCNFT50 revealed good cell attachment after seven days attributed to the porous structure and major number of sites for cell adhesion, which was related to excellent cytocompatibility and the porous network structure that resembles the natural extra-cellular matrix (ECM), and the outer alginate chains containing hydrophilic characteristics that can facilitate cell attachment [35,67].

Hence, the present in vitro results demonstrated that the AlgCNFT50 gels present good cytocompatibility and have potential applications in tissue engineering owing to their referred biocompatibility, potential low cost, and renewable character of the raw materials.

## 4. Conclusions

In this work, different nanocellulose-alginate hydrogels containing cellulose nanocrystals (CNCs), TEMPO-oxidized CNCs (CNCTs), cellulose nanofibers (CNFs) or TEMPO-oxidized CNFs (CNFTs) at different concentrations (10, 36, and 50 wt %) were synthetized for biomedical applications. Differently from the neat alginate gels, with pore diameters in the range of 20 up to 40 µm, the gels containing nanocelluloses presented the formation of pores with diameters ranging from 40 to 150 μm, which are more suitable for the transport of nutrients and cell growth. Dynamical mechanical thermal analyses showed that a thermal post-treatment plays a crucial role on the mechanical properties of the samples due to an increase in the Ca^2+^ crosslinking density within the gels. Also, the gels containing CNFTs showed an increase of storage modulus with an increase of nanocellulose contents in the second cycle of heating, due to the presence of carboxylic groups on nanofibers that allows for a better thermally-induced interaction between alginate and CNFTs through ionic crosslinking. As hypothesized, the thermal post-treatment plays an important role on the physicochemical properties of the gels and the materials containing TEMPO-oxidized cellulose nanofibers (AlgCNFT50). They showed the best compatibility conditions for bioadhesion and growth of fibroblasts cells on their surface due to the presence of an entangled nanofiber network which ensured their dimensional stability, good mechanical properties, and hydrophilic characteristics. Thus, the alginate-nanocellulose gels prepared in this work can be considered as interesting materials which are able to resemble the natural extra-cellular matrix (ECM) of tissues. More aspects on biological properties and potential biomedical applications of this material need to be examined in future works and before in vivo tests. However, the preliminary results obtained here including mechanical properties, morphology, cytocompatibility, bioadhesion, and cell proliferation suggest that this material can be a potential candidate to find applications in tissue engineering area as, for instance, in tissue repair and wound healing.

## Figures and Tables

**Figure 1 nanomaterials-09-00078-f001:**
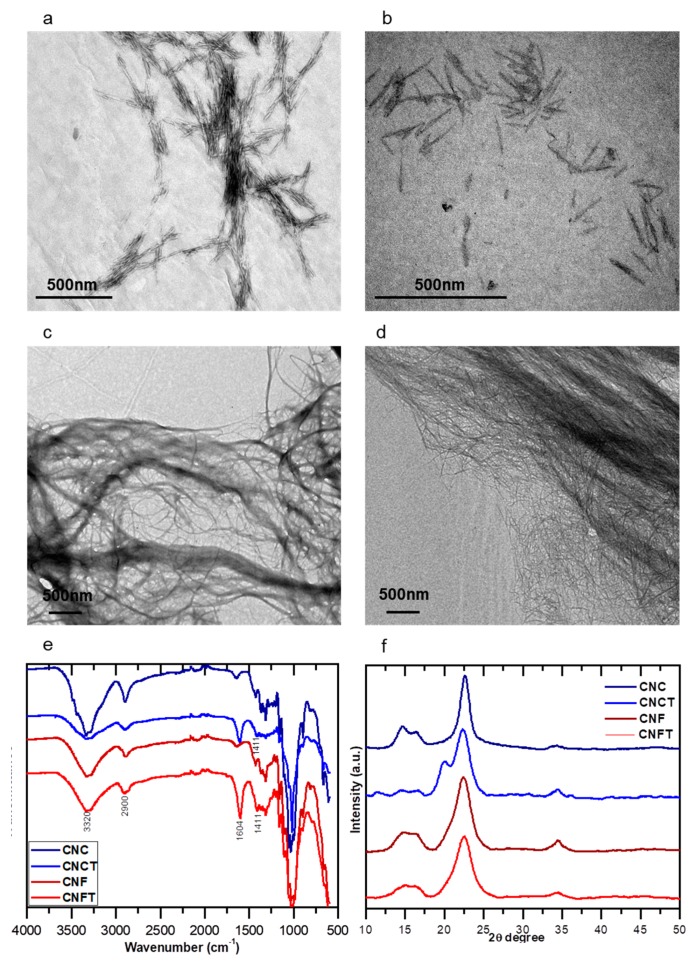
Morphological and structural characteristics of the unmodified and oxidized nanocellulose samples. Transmission electron micrographs (TEM) of: (**a**) cellulose nanocrystals (CNCs); (**b**) TEMPO-oxidized cellulose nanocrystals (CNCTs); (**c**) cellulose nanofibers (CNFs); and (**d**) TEMPO-oxidized cellulose nanofibers (CNFT). FTIR spectra of CNC, CNCT, CNF and CNFT (**e**) and (**f**) X-ray diffraction (XRD) patterns of CNC, CNCT, CNF, and CNFT.

**Figure 2 nanomaterials-09-00078-f002:**
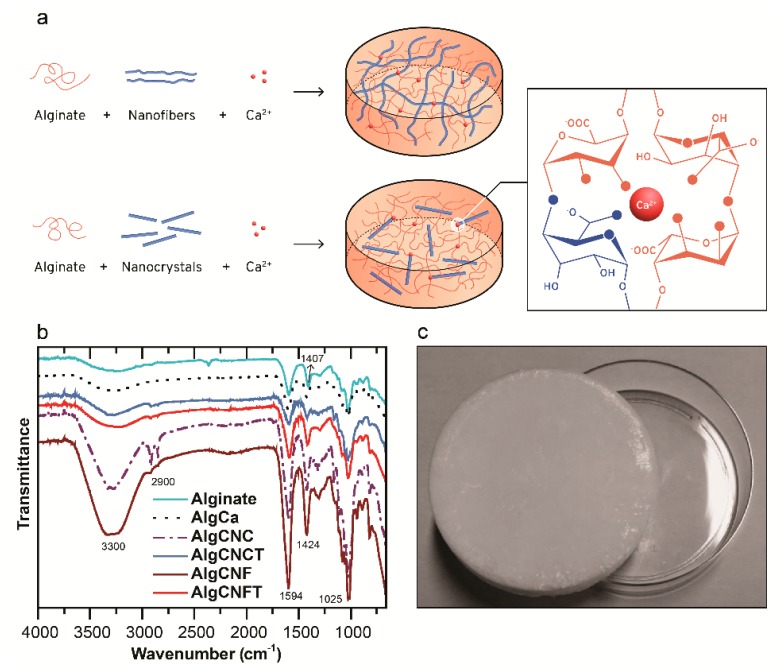
Formulation and characterization of nanocellulose-alginate based hydrogels. (**a**) Schematic representation of the crosslinking reaction of alginate gels and TEMPO-oxidized nanocellulose with Ca^2+^ ions. (**b**) FTIR spectra of the pure sodium alginate polymer (starting material), the pure alginate gel (AlgCa) and the nanocellulose alginate gels at 36 wt % of nanocelluloses (AlgCNC36, AlgCNCT36, AlgCNF36, AlgCNFT36). (**c**) Digital image of the gel AlgCNFT36.

**Figure 3 nanomaterials-09-00078-f003:**
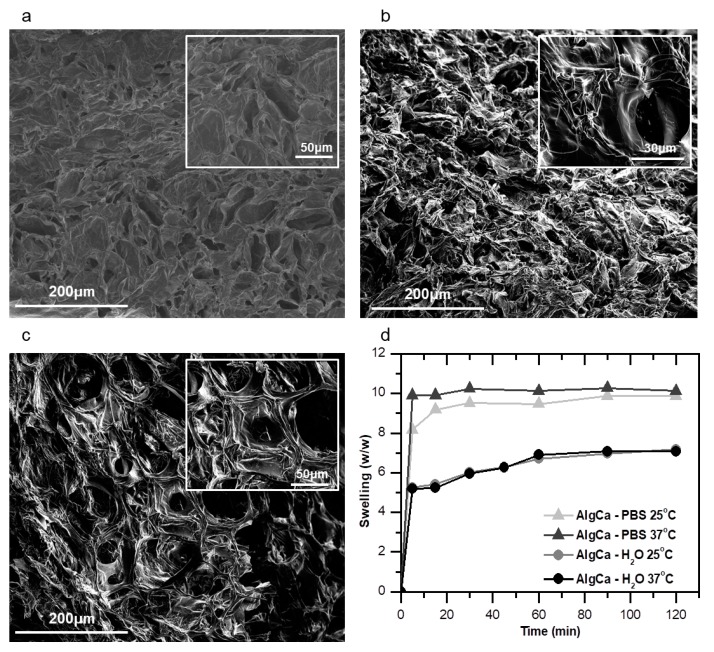
SEM images of the cross-section of crosslinked gels. (**a**) Pure alginate (AlgCa); (**b**) alginate-CNCT gel at 50 wt %; and (**c**) alginate-CNFT gel at 50 wt % of nanocelluloses; 500× magnification (**a**–**c**). The insets represent magnification of 1500× (**a**,**c**) and 3500× (**b**); (**d**) swelling kinetics of the calcium crosslinked alginate-gel (AlgCa).

**Figure 4 nanomaterials-09-00078-f004:**
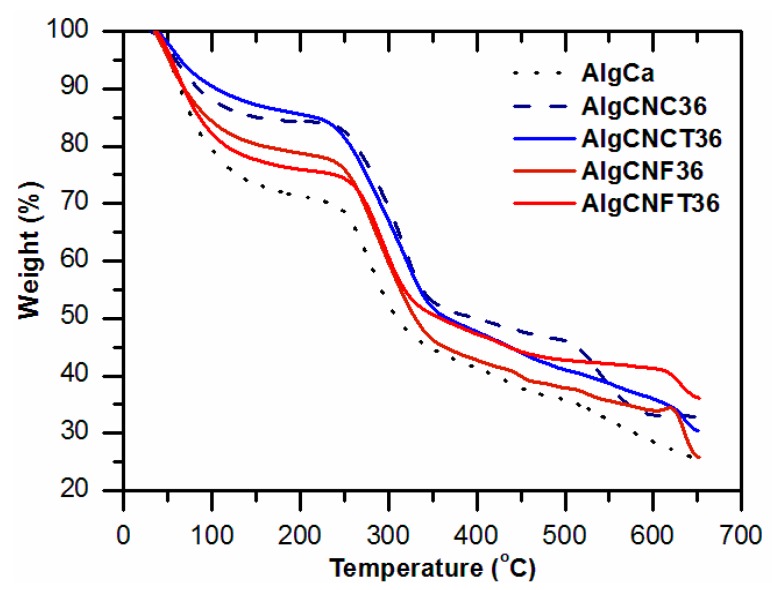
Thermogravimetric analyses of pure alginate and nanocellulose-alginate gels. Influence of the different nanocellulose types (CNC, CNCT, CNF, CNFT, at 36 wt %) on the thermal stability of the alginate gels.

**Figure 5 nanomaterials-09-00078-f005:**
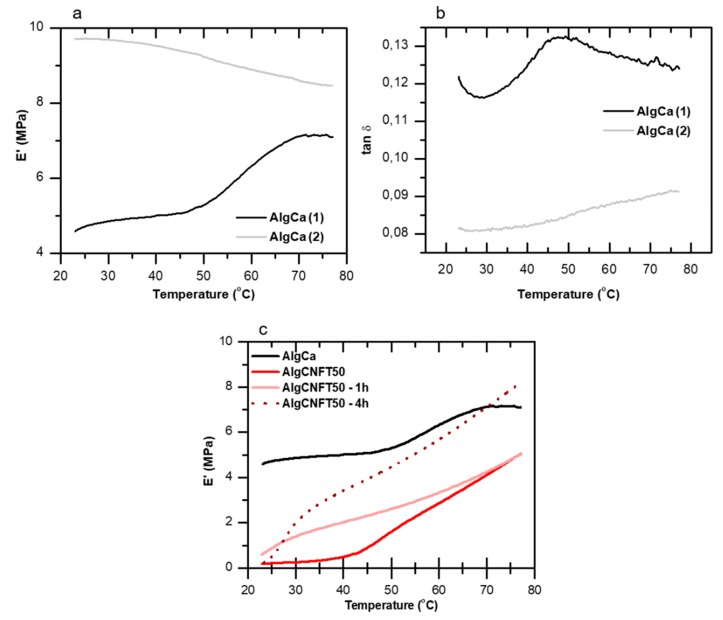
Dynamic mechanical thermal analyses of alginate-gels. Changes of (**a**) storage moduli (E’) and (**b**) tan δ for alginate gel without nanocelluloses addition (AlgCa) during two heating cycles. (**c**) Influence of thermal post-treatment on the samples carried out in an oven at 60 °C for 0, 1 or 4 h for alginate gels at 50 wt % concentration of TEMPO-oxidized cellulose nanofibers (AlgCNFT).

**Figure 6 nanomaterials-09-00078-f006:**
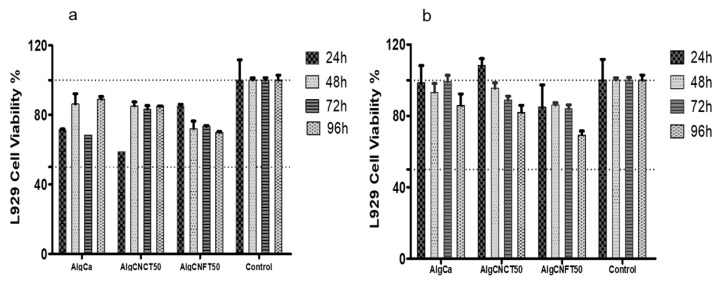
L929-fibroblast cells 3-(4,5-dimethylthiazolyl-2)-2,5-diphenyltetrazolium bromide (MTT) assays. (**a**) Direct contact assay cytotoxicity and cell proliferation were evaluated for 24, 48, 72, and 96 h. (**b**) Indirect contact assay (eluate for 48 h) cytotoxicity and cell proliferation were evaluated for 24, 48, 72, and 96 h. Data represent the mean ± standard deviation of six replicates for each material (*n* = 6).

**Figure 7 nanomaterials-09-00078-f007:**
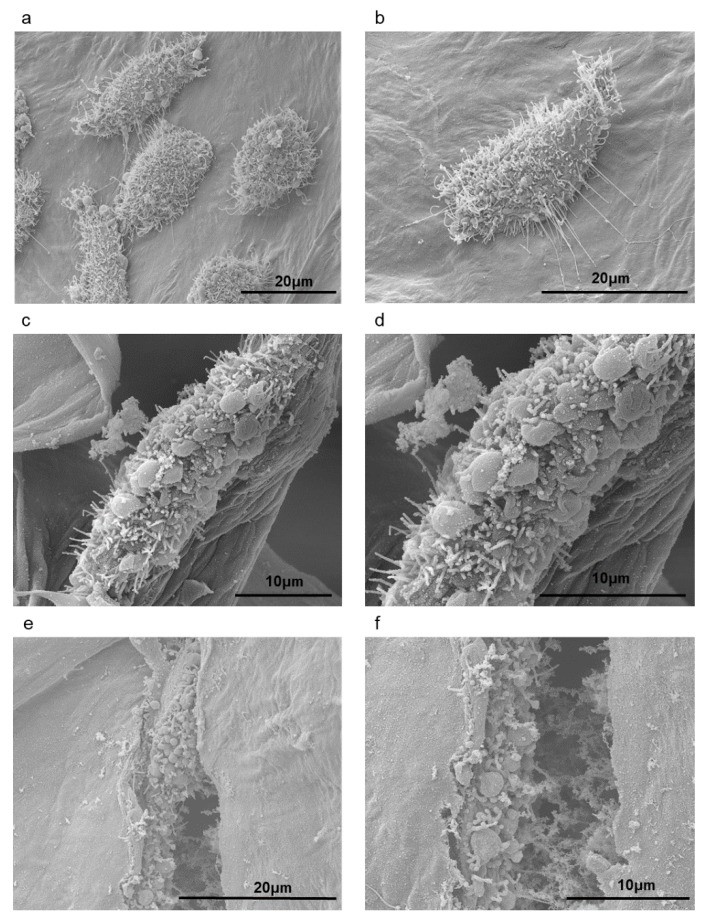
SEM micrographs of L929-fibroblast cells seeding on AlgCNFT50 gels: 2 days of incubation at 4000× (**a**) and 6000× magnification (**b**); 4 days of incubation at 4000× (**c**) and 6000× magnification (**d**); 7 days of incubation at 6000× (**e**) and 10,000× magnification (**f**).

**Table 1 nanomaterials-09-00078-t001:** Composition of the reagents used in the nanocellulose-alginate gels preparation.

Sample Name	Alginate (g)	Nanocellulose (g)	Nanocellulose/Alginate Weight Ratio
**AlgCa**	0.200	0.000	0.000
**AlgCNC10**	0.281	0.030	0.100
**AlgCNC36**	0.200	0.112	0.560
**AlgCNC50**	0.156	0.156	1.000
**AlgCNCT10**	0.281	0.030	0.100
**AlgCNCT36**	0.200	0.112	0.560
**AlgCNCT50**	0.156	0.156	1.000
**AlgCNF10**	0.281	0.030	0.100
**AlgCNF36**	0.200	0.112	0.560
**AlgCNF50**	0.156	0.156	1.000
**AlgCNFT10**	0.281	0.030	0.100
**AlgCNFT36**	0.200	0.112	0.560
**AlgCNFT50**	0.156	0.156	1.000

**Table 2 nanomaterials-09-00078-t002:** Maximum moisture uptake after swelling tests during 2 h at 37 °C in milli-Q water pH 6.2 and in PBS pH 7.4 for the pure alginate gels and alginate-nanocellulose gels.

Sample	Moisture Uptake (g/g) Milli-Q H_2_O; pH 6.2	Moisture Uptake (g/g) PBS; pH 7.4
**AlgCa**	6.64 ± 0.05	10.75 ± 0.62
**AlgCNC10**	5.86 ± 0.20	9.16 ± 0.91
**AlgCNC36**	8.33 ± 0.05	8.52 ± 0.59
**AlgCNC50**	7.47 ± 0.46	8.89 ± 1.95
**AlgCNCT10**	7.57 ± 0.77	7.35 ± 0.68
**AlgCNCT36**	8.44 ± 0.45	9.01 ± 1.28
**AlgCNCT50**	8.24 ± 0.51	11.45 ± 0.50
**AlgCNF10**	6.86 ± 0.13	11.45 ± 0.42
**AlgCNF36**	9.78 ± 0.10	11.50 ± 0.56
**AlgCNF50**	11.36 ± 0.61	13.19 ± 2.00
**AlgCNFT10**	16. 31 ± 0.58	17.34 ± 0.85
**AlgCNFT36**	17.73 ± 0.78	17.72 ± 0.86
**AlgCNFT50**	14.61 ± 0.06	18.13 ± 0.56

**Table 3 nanomaterials-09-00078-t003:** Storage modulus after first (E’_1_) and second (E’_2_) heating/cooling cycles of the gels at 35 and 50 wt % nanocelluloses concentration and in the range of human body temperature (35, 37, and 42 °C).

SAMPLE	35 °C		37 °C		42 °C	
	E’_1_ (MPa)	E’_2_ (MPa)	E’_1_ (MPa)	E’_2_ (MPa)	E’_1_ (MPa)	E’_2_ (MPa)
**AlgCNC36**	2.27	2.12	2.67	2.63	3.00	3.02
**AlgCNC50**	0.10	0.12	0.10	0.11	0.10	0.12
**AlgCNCT36**	4.49	4.48	5.25	5.24	7.83	7.81
**AlgCNCT50**	0.97	0.96	1.14	1.12	1.22	1.21
**AlgCNF36**	0.15	2.13	0.16	2.10	0.47	2.29
**AlgCNF50**	0.02	0.23	0.02	0.24	0.02	0.22
**AlgCNFT36**	1.13	2.52	1.30	2.55	1.66	2.66
**AlgCNFT50**	0.33	3.46	0.40	3.54	0.62	3.74

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
