# Peer review of "Three-Dimensional Stable Alginate-Nanocellulose Gels for Biomedical Applications: Towards Tunable Mechanical Properties and Cell Growing"

_nanomaterials, 2019, doi:10.3390/nano9010078_

Round 1
Reviewer 1 Report
Here authors report on the fabrication of alginate hydrogels reinforced with CNCs and CNFs. These nanocelluloses were also modified following the well-known TEMPO-mediate oxidation to understand the effect of surface treatment on the resulting properties of hydrogels. The work is carefully done, well written and results are well presented. I estimate it may be of interest for the potential readers of the Journal, so I suggest its acceptance for publication after some minor issues are corrected:
· Authors state that “functionalized nanocelluloses can influence the final properties of the gels since the nanocelluloses are able to change their surface charges”. However, no surface charge determination has been done. Authors should run Zeta-potential measurements on diverse nanocelluloses as this information may help to provide further insights on the interaction between nanocelluloses and the matrix.
· I suggest enlarging XRD patterns in the 2θ 10-50° region to better show the most relevant section. I also think the quality if the patterns should be improved as currently it seems too noisy.
· As notable differences are induced, authors must report a pore-size distribution graph for different hydrogels. This may serve to better explain the increase on the moisture uptake upon nanocellulose addition.
· For the sake of comparison, I think authors should report DMA results for alginate and CNC/CNF reinforced alginate hydrogels in the same graph as shown in Figure 5a and b. This may help the reader to understand this section.
Author Response
Attached please find the author reply to Reviewer #1

Reviewer 2 Report
This study is about 3D alginate-nanocellulose gels for biomedical applications. The nanocelluloses included in the study are CNC and CNF, and the corresponding carboxylated grades. The study is comprehensive and provides additional data regarding the applicability of nanocellulose for biomedical use.
Some specific comments for consideration are:
Line 60: please provide a reference to this statement about low cell attachment.
Line 118: specify the type of Eucalyptus that was used.
Line 123: why was this alginate chosen for this study when the M/G was >1? Alginates with M/G<1 form stronger gels and the mechanical performance was an important characteristic in this study.
Line 138: Masuko grinder may release particles due to friction of the grinding stones. Are these potential particles a limiting factor when producing nanocellulose for biomedical applications? Please comment in the text.
Line 149: Is the eucalyptus pulp the same as the one used for CNF production?
Line 175: A degree of oxidation of 10 mmol/g is extremely high, why was this chosen? This doesn’t correspond to the values reported in Line 335.
Section 3.1: The authors focus on the nanoscale dimensions of the nanocelluloses and provide some average dimensions, please specify how many measurements were undertaken and were the measurements performed randomly? From how many images? These materials contain probably residual fibres, specially the CNF and probably the CNFT. This is a structural feature that is neglected and needs to be reported in the study.
Line 350: Calcium can also be extracted from CNFT networks and probably more easily due to the lower Ca-CNFT interactions, compared to alginates. Please comment.
Line 465: This statement about SSA cannot be justified. The SEM images don’t show a clear difference between the structures. Please modify this sentence. This applies to the whole discussion about the differences observed in the SEM images.
Line 603-606: There are specific ISO standards for evaluating the cytotoxicity of materials for biomedical devices. See Basu et al. 2017. Cell viability below 70% indicates that the tested material is cytotoxic. Please modify this part of the results and discussion.
Additionally, there are some typos and grammatical errors that require attention.
Author Response
Attached please find the author reply to Reviewer #2
